# Dynamic and Static ^18^F-FDG PET/CT Imaging in SMARCA4-Deficient Non-Small Cell Lung Cancer and Response to Therapy: A Case Report

**DOI:** 10.3390/diagnostics13122048

**Published:** 2023-06-13

**Authors:** Xieraili Wumener, Xiaoxing Ye, Yarong Zhang, Shi Jin, Ying Liang

**Affiliations:** 1Department of Nuclear Medicine, National Cancer Center/National Clinical Research Center for Cancer/Cancer Hospital & Shenzhen Hospital, Chinese Academy of Medical Sciences and Peking Union Medical College/Shenzhen Clinical Research Center for Cancer, Shenzhen 518116, China; xieraili473@163.com (X.W.); zhangyarong_123@163.com (Y.Z.); 2Department of Pathology, National Cancer Center/National Clinical Research Center for Cancer/Cancer Hospital & Shenzhen Hospital, Chinese Academy of Medical Sciences and Peking Union Medical College/Shenzhen Clinical Research Center for Cancer, Shenzhen 518116, China; pathologyyxx@163.com; 3Department of Medical Oncology, National Cancer Center/National Clinical Research Center for Cancer/Cancer Hospital & Shenzhen Hospital, Chinese Academy of Medical Sciences and Peking Union Medical College/Shenzhen Clinical Research Center for Cancer, Shenzhen 518116, China

**Keywords:** ^18^F-FDG, PET/CT, SMARCA 4, NSCLC, dynamic imaging, efficacy evaluation

## Abstract

SMARCA4-deficient non-small cell lung cancer (NSCLC) is a more recently recognized subset of NSCLC. We describe the ^18^F-fluorodeoxyglucose (FDG) PET/CT findings in a rare case of SMARCA4-deficient NSCLC and response to therapy. A 45-year-old male patient with a history of heavy smoking (10 years) underwent an ^18^F-fluorodeoxyglucose (FDG) PET/CT dynamic (chest) + static (whole-body) scan for diagnosis and pre-treatment staging. ^18^F-FDG PET/CT showed an FDG-avid mass in the upper lobe of the left lung (SUV_max_ of 22.4) and FDG-avid lymph nodes (LN) in the left pulmonary hilar region (SUV_max_ of 5.7). In addition, there were multiple metastases throughout the body, including in the distant LNs, adrenal glands, bone, left subcutaneous lumbar region, and brain. Pathological findings confirmed SMARCA4-deficient NSCLC. After four cycles of chemotherapy and immune checkpoint inhibitors (ICI), the patient underwent again an ^18^F-FDG PET/CT scan (including a dynamic scan) for efficacy evaluation. We report a case that deepens the understanding of the ^18^F-FDG PET/CT presentation of SMARCA4-deficient NSCLC as well as dynamic imaging features and parametric characteristics.

SMARCA4-deficient non-small cell lung cancer (NSCLC) accounts for 3–6% of all NSCLCs [1]. It is only in recent years has emerged as a distinct NSCLC subset [2]. SMARCA4-deficient NSCLC is prevalent in men aged 40–50 years and shows a strong association with smoking [3]. Deficient SMARCA4 leads to an increased incidence of tumor metastasis [4]. It is highly aggressive, rapidly progressive, and has a poor prognosis [1,4,5,6]. Previous studies have reported that approximately 83% of SMARCA4-deficient NSCLCs are already in stage IV at the time of detection and have a progression-free survival of only 30 days [4,7]. There are very limited descriptions of SMARCA4-deficient NSCLC morphology as well as ^18^F-FDG PET/CT features. To our knowledge, we are the first to present a dynamic imaging and metabolic parameter profile for SMARCA4-deficient NSCLC.

Effective treatment for SMARCA4-deficient NSCLC has not been established. Little is known about the efficacy of ICIs in SMARCA4-deficient NSCLC. In 2019, Tomoyuki Naito et al. reported the first case of SMARCA4-deficient NSCLC successfully treated with nivolumab [8]. In the case, we reported, based on the ^18^F-FDG PET/CT findings (Figure 1), the patient’s pathological and immunohistochemical findings confirmed SMARCA4-deficient NSCLC (Figure 2), and clinical stage was T2N3M1, stage IVB. Then the patient received four cycles of ICIs (Tislelizumab, 200 mg) and chemotherapy (Paclitaxel 0.2 g + Carboplatin 450 mg). The review ^18^F-FDG PET/CT after four cycles of treatment (4 months later, April 2023, Figure 3) showed that the primary lung cancer foci, distant metastases, and retroperitoneal LN were significantly smaller than before, and the SUV_max_ was reduced. Other than that, tumor markers reviewed regularly during treatment were negative. However, the MRI (brain, follow up to March 2023) results at follow-up showed that the brain metastases did not shrink significantly, and enhanced on enhancement scans. Therefore, the clinical efficacy evaluation after four cycles of treatment was rated as stable disease (SD). As of our last follow-up (May 2023), the patient was on cycle 6 of treatment. So far, the general condition is good and no recurrent/metastatic lesions have been observed. We will follow up further on the patient’s condition.

Dynamic ^18^F-FDG PET/CT (dPET/CT) extracts physiological parameters which can better reveal the pathophysiological mechanisms of diseases. Many previous studies have confirmed that the K_i_ values of malignant lesions are higher than those of benign lesions [9,10,11]. Our previous study also concluded that in lung cancer, the K_i_ of metastatic LNs was higher than that of non-metastatic LNs (0.019 vs. 0.016/mLg/min, *p* = 0.001) [11]. However, studies evaluating the efficacy of ICIs in lung cancer were seldom seen. In this case, the enlarged LNs in the mediastinum and pulmonary hilar region after treatment raised our attention, and these LNs are more clearly seen from the K_i_ images (Figure 3E). Therefore, we studied the pre-treatment and post-treatment ratios of static and dynamic parameters (Table 1) for chest lesions (including the upper lobe of the left lung, LN in the left pulmonary hilar region, and the right eighth rib). The results showed that both the primary left lung cancer focus (ΔSUV_max_ of 90.18%, and ΔK_i_ of 85.52%) and the right eighth rib ratio were high (ΔSUV_max_ of 70.21%, and ΔK_i_ of 68.22%), but the left hilar LN ΔSUV_max_ was low (29.83%) and ΔK_i_ was high (74.03%), and the trend of ΔK_i_ remained consistent with the primary focus. As the patient did not undergo further puncture biopsy for enlarged LNs after the second ^18^F-FDG PET/CT scan. It is not known at this time whether these enlarged LNs are metastatic LNs or benign LNs associated with ICI treatment. Therefore, we propose to hypothesize whether dynamic metabolic parameters (K_i_) could provide additional metabolic information for the assessment of ICI efficacy. This deserves further investigation and is something we are studying.

## Figures and Tables

**Figure 1 diagnostics-13-02048-f001:**
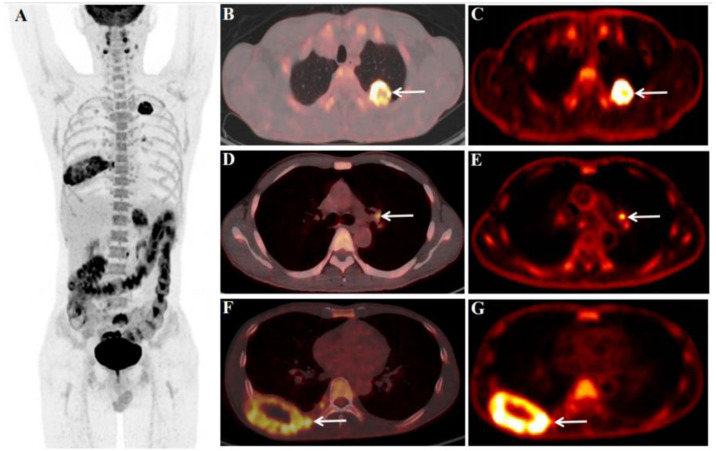
**Initial scan:** We report the case of a 45-year-old male patient with a history of heavy smoking (10 years, 20 cigarettes/day) underwent an ^18^F-fluorodeoxyglucose (FDG) PET/CT dynamic (chest, **C**,**E**,**G**) + static (whole-body, **A**,**B**,**D**,**F**) scan (December 2022) for diagnosis and pre-treatment staging. The abnormal serum tumor markers associated with lung cancer before ^18^F-FDG PET/CT were CYFRA21-1 18.57 ng/mL (<3.3) and CA125 64.60 U/mL (<35.0). Before the ^18^F-FDG injection, the patient had fasted for at least 6 h and had a pre-scan glucose level of 4.4 mmol/L. According to the body mass index, the chest region PET scans (dynamic) were initiated immediately after the injection of ^18^F-FDG (7.07 mCi) from an intravenous indwelling needle. The total dynamic scans lasted for 65 min. Dynamic scan data were partitioned into 28 frames as follows: 6 × 10 s, 4 × 30 s, 4 × 60 s, 4 × 120 s, and 10 × 300 s. An additional whole-body static PET/CT scan was performed at the end of the dynamic acquisition. Quantitative parameters (K_i_) were obtained through applying the irreversible two-tissue compartment model using in-house Matlab software. ^18^F-FDG PET/CT showed an FDG-avid mass in the upper lobe of the left lung (**B**,**C**, white arrow), size of 3.4 × 3.0 cm, SUV_max_ of 22.4 (**B**), K_i_ of 0.0525 /mLg/min (**C**), FDG-avid LN in the left pulmonary hilar region (**D**,**E**, white arrow), size of 1.1 × 0.9 cm, SUV_max_ of 5.7 (**D**), K_i_ of 0.0231/mLg/min (**E**), and bone destruction with the mass formation in the right eighth rib (**F**,**G**, white arrow), size of 8.0 × 4.5 cm, SUV_max_ of 9.4 (**F**), K_i_ of 0.0214 /mLg/min (**G**). Multiple metastases throughout the body (**A**), including retroperitoneal LN (size of 1.3 × 1.0 cm, SUV_max_ of 8.3), left adrenal gland (size of 3.8 × 3.5 cm, SUV_max_ of 8.4), bone of the sacrum (size of 3.6 × 3.1 cm, SUV_max_ of 10.9), left subcutaneous lumbar region (size of 1.7 × 1.3 cm, SUV_max_ of 7.6), and brain (size of 1.9 × 1.8 cm, SUV_max_ of 9.1).

**Figure 2 diagnostics-13-02048-f002:**
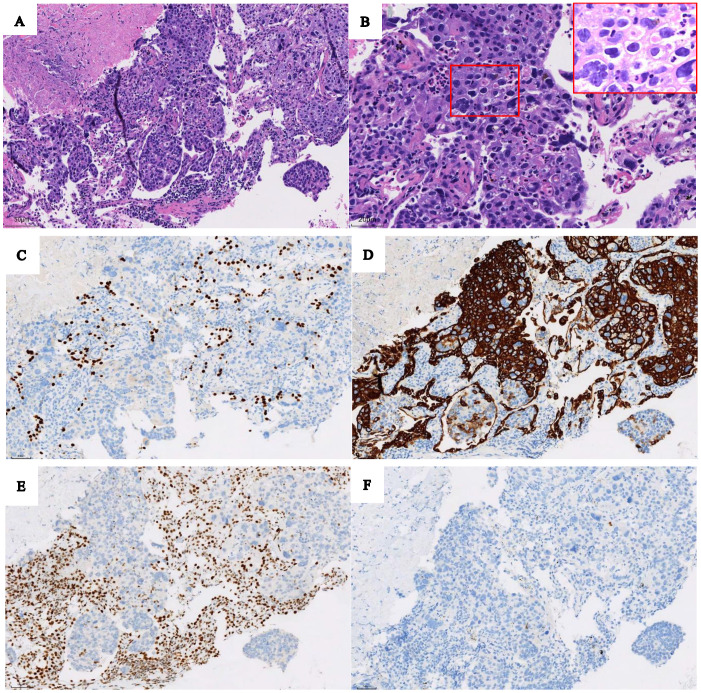
Based on the ^18^F-FDG PET/CT findings, the patient underwent a puncture biopsy of a mass in the upper lobe of the left lung. Pathological and immunohistochemical findings confirmed SMARCA4-deficient NSCLC. Hematoxylin-eosin staining (original magnification ×200 (**A**)) showed the tumor cells arranged in sheets and lobules, and ((**B**) original magnifications ×400) showed sheets of large pleomorphic tumor cells with prominent nucleoli. At immunohistochemistry (**C**–**F**, original magnification ×200), tumor cells were not express TTF1 (**C**) with alveolar epithelial cells as internal positive controls; diffuse expression of cytokeratin 7 (**D**) was seen in tumor cells, and loss of BRG1 protein (**E**) in tumor cells with mesenchymal cells and inflammatory cells serving as internal positive controls, and P63 (**F**) was a negative expression in tumor cells.

**Figure 3 diagnostics-13-02048-f003:**
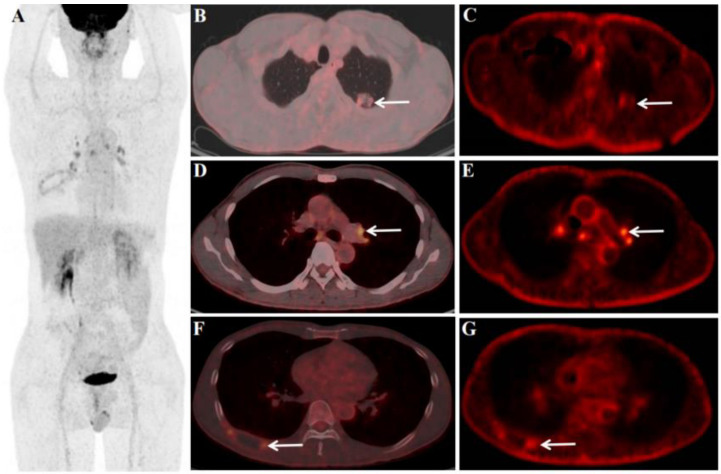
**Follow-up:** After four cycles of immune checkpoint inhibitors (ICIs) and chemotherapy (Paclitaxel + Carboplatin), the patient underwent an ^18^F-FDG PET/CT scan (**A**–**G**, including dynamic scan, April 2023); the aim is to perform an efficacy evaluation. After treatment, ^18^F-FDG PET/CT dynamic + static scan showed left lung primary foci were smaller than before and FDG uptake was reduced, size of 2.1 × 1.2 cm, SUV_max_ of 2.2 (**B**), K_i_ of 0.0070 /mLg/min (**C**), ΔSUV_max_ (pre-treatment SUV_max_ − treatment SUV_max_/pre-treatment SUV_max_) of 90.18%, and ΔK_i_ (pre-treatment K_i_ − treatment K_i_/pre-treatment K_i_) of 86.67%. For the LNs, most of the LNs in the mediastinal region and pulmonary hilar regions were larger and FDG-avid than before treatment. Among them, although the left pulmonary hilar LN was larger than before, the FDG uptake and K_i_ were reduced than before, size of 1.2 × 0.8 cm, SUV_max_ of 4.0 (**D**), K_i_ of 0.0060 /mLg/min (**E**), ΔSUV_max_ of 29.83%, and ΔK_i_ of 74.03%. The right eighth rib was also significantly smaller than before treatment, and FDG uptake was reduced, size of 4.5 × 1.8 cm, SUV_max_ of 2.8 (**F**), K_i_ of 0.0068 /mLg/min (**G**), ΔSUV_max_ of 70.21%, and ΔK_i_ of 68.22%. In addition, distant metastases were smaller than before and FDG uptake was reduced, including the retroperitoneal LN (not clearly shown), left adrenal gland (size of 1.8 × 1.2 cm, SUV_max_ of 2.1, ΔSUV_max_ of 75.0%), bone of the sacrum (size of 3.5 × 3.0 cm, SUV_max_ of 2.1, ΔSUV_max_ of 80.73%), left subcutaneous lumbar region (size of 0.9 × 0.8 cm, SUV_max_ of 1.4, ΔSUV_max_ of 81.58%), and brain (size of 1.0 × 0.8 cm, not uptake seen).

**Table 1 diagnostics-13-02048-t001:** Characteristics of dynamic and static ^18^F-FDG PET/CT imaging and metabolic parameters before and after treatment. ^1^: Highest FDG uptake LN. ^2^: Pre-treatment SUV_max_ − treatment SUV_max_/pre-treatment SUV_max_. ^3^: Pre-treatment K_i_ − treatment K_i_/pre-treatment K_i_.

Section	Before Treatment			After Treatment			Parameter Change	
	Size	SUV_max_	K_i_	Size	SUV_max_	K_i_	ΔSUV_max_ ^2^	ΔK_i_ ^3^
Upper lobe of left lung	3.4 × 3.0	22.4	0.0525	2.1 × 1.2	2.2	0.0070	90.18%	86.67%
Mediastinal LN	-	-	-	1.3 × 1.1 (zone 7) ^1^	3.8	0.0064	-	-
Pulmonary hilar LN	1.1 × 0.9 (left)	5.7	0.0231	1.2 × 0.8 (left)	4.0	0.0060	29.83%	74.03%
	-	-	-	1.3 × 1.2 (right) ^1^	3.9	-	-	-
Retroperitoneal LN	1.3 × 1.0	8.3	-	Not clearly shown	-	-	-	-
Left adrenal gland	3.8 × 3.5	8.4	-	1.8 × 1.2	2.1	-	75.0%	-
Brain	1.9 × 1.8	9.1	-	1.0 × 0.8	No uptake seen	-	-	-
Right 8th rib	8.0 × 4.5	9.4	0.0214	4.5 × 1.8	2.8	0.0068	70.21%	68.22%
Bone of the sacrum	3.6 × 3.1	10.9	-	3.5 × 3.0	2.1	-	80.73%	-
Left subcutaneous lumbar region	1.7 × 1.3	7.6	-	0.9 × 0.8	1.4	-	81.58%	-

## Data Availability

Not applicable.

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
