# Peer review of "Dynamic and Static 18F-FDG PET/CT Imaging in SMARCA4-Deficient Non-Small Cell Lung Cancer and Response to Therapy: A Case Report"

_diagnostics, 2023, doi:10.3390/diagnostics13122048_

Round 1

Reviewer 1 Report

Review of the paper titled: “Dynamic and static 18F-FDG PET/CT imaging in SMARCA4-deficient 2 non-small cell lung cancer and response to therapy: A Rate Case” by Xieraili Wumener et al

Thank you for giving me the chance to review this interesting paper. 

Somatic mutation in tumour suppressor gene SMARCA4 (SWI/SNF–related, matrix-associated, actin-dependent regulator of chromatin, subfamily A, member 4) identifies a subset of NSCLC lacking alterations in EGFR , ALK and ROS1 genes; this kind of mutation damages of transcriptional processes regulation, cellular differentiation and DNA injury repair. 

First reported in 2000, SMARCA4 homozygous mutations/deletions have been identified in Approximately 10% of non-small cell lung cancers. 

Loss of SMARCA4 leads to the occurrence of advanced dedifferentiated tumors and increases the incidence of tumor metastasis. Approximately 83% of patients with SMARCA4-deficient non-small cell lung cancer (SMARCA4-dNSCLC) are at stage IV at the time of discovery, with a median progression-free survival time of only 30 days.

SMARCA4 loss causes responsiveness to immunotherapy and targeted therapies. 

Platinum-based chemotherapy regimens combined with immune checkpoint inhibitors) are currently most commonly used in clinical practice but their efficacy is unknown. Nevertheless, it is known that some patients can benefit from first-line immunotherapy. Therefore, there is a need to clearly classify NSCLC subtypes by accurate molecular testing to understand the evolution of the condition and select a reasonable treatment modality. 

The authors describe the clinic and pathological features of n=1 SMARCA4-dNSCLC patient. The report is interesting, and figures are of sufficient quality. 

Case description needs more information about follow-up (timing between the first and second PET scan), patient survival; even chemotherapy regimen should be more detailed. 

English language is good

Author Response

Many thanks to the you for their professional comments.

    Based on your comments, we have added detailed information about the follow-up in the corresponding paragraphs, such as the timing of the first and second PET/CT scans, chemotherapy regimen, patient survival information, clinical staging, etc. In addition, thank you for providing the relevant expertise points. We have identified and studied the relevant literature and added the relevant knowledge points.

    Thank you again for your professional guidance.

Reviewer 2 Report

 Post treatment results must be represented

1. It must be well-described what is the novel outcome of this work for 18F-FDG PET/CT. 2. Pre and post treatment 18F-FDG PET/CT images must be presented and compared. 3. Pre and post treatment histological images must be presented and compared. 4. The method must be described in details.

Overall, quality of English is fine while minor improvements can be considered.

Author Response

Thank you very much for your professional guidance.

1) The 18F-FDG PET/CT images and imaging features before and after treatment in this case were refined and supplemented accordingly, as shown in Table 1.

2) The section on methods has been mentioned in the manuscript.

3) This patient was diagnosed pathologically after PET/CT examination only. As of the current position, no histopathological confirmation (including a second PET/CT) was done on the patient during the follow-up. This section has been added to the revised manuscript.

4) In this case report, we have proposed a hypothesis: whether dynamic quantitative metabolic parameters could bring more metabolic information in the assessment of immune checkpoint inhibitor efficacy in lung cancer and provide reliable metabolic information for clinical purposes. This hypothesis needs to be studied and validated with larger sample size and studies. This is the study we are currently conducting.

Thank you again for your professional guidance.

Reviewer 3 Report

Dear Authors! 

Please make the results more detailed. 

Use a paragraph before each figure to explain the case and the experiment.

It is quite good.

Author Response

Thank you very much for your guidance.

  1. In the revised version, we have described and added details to the results.
  2. Since this manuscript was submitted to the "Interesting Images" section. Therefore, we have included the relevant content after the image, based on previous reports in the journal.

 Thank you again for your professional guidance.

Reviewer 4 Report

The article ‘Dynamic and static 18F-FDG PET/CT imaging in SMARCA4-deficient non-small cell lung cancer and response to therapy: A Rate Case’ submitted by Wumener X. et al. describes a case study where authors have used 18F-fluorodeoxyglucose (FDG) PET/CT imaging for the diagnosis of SMARCA4-deficient NSCLC (non-small cell lung cancer) of a 45yr old patient with a history of heavy smoking. According to the authors, the patient underwent 18F-fluorodeoxyglucose (FDG) PET/CT scan for diagnosis, pre-treatment and post-treatment. 18F-fluorodeoxyglucose (FDG) PET/CT imaging is a powerful imaging tool for detection of cancers in cancer patients.

I have a few concerns, which I have mentioned here-

1.     Authors should make a ‘Results/Discussion’ section after the introduction and discuss their findings. Authors have presented three figures with elaborated figure legends which should have been covered in the ‘Results/Discussion’ section.

2.     Is it the first case where anyone used 18F-fluorodeoxyglucose (FDG) PET/CT to diagnose SMARCA4-deficient non-small cell lung cancer?

Author Response

Thank you very much for taking the time to give us your professional guidance.

  1. We submitted this case in the "Interesting Images" section. Therefore, according to the previous reports in the journal, we did not have a discussion section in the manuscript. However, we have included the relevant content after the images.
  2. According to our collection of relevant studies, relevant PET/CT has been reported in the past, with a predominance of static imaging features. We were fortunate to collect this case in our study on dynamic imaging features of lung cancer. After reviewing the relevant literature, we found no previous reports of dynamic PET/CT on SMARCA-4 deficient non-small cell lung cancer. Therefore, we have written this case report for reference and explored the characteristics of dynamic metabolic parameters. We hope you will give us more professional guidance.

  Thank you again for your professional guidance.

Round 2

Reviewer 2 Report

Comments are applied and now it looks that enough information is provided.